# PURCHASE AS REWARD: SESSION-BASED RECOMMENDATION BY IMAGINATION RECONSTRUCTION

## ABSTRACT

One of the key challenges of session-based recommender systems is to enhance users' purchase intentions. In this paper, we formulate the sequential interactions between user sessions and a recommender agent as a Markov Decision Process (MDP). In practice, the purchase reward is delayed and sparse, and may be buried by clicks, making it an impoverished signal for policy learning. Inspired by the prediction error minimization (PEM) and embodied cognition, we propose a simple architecture to augment reward, namely Imagination Reconstruction Network (IRN). Specifically, IRN enables the agent to explore its environment and learn predictive representations via three key components. The *imagination core* generates predicted trajectories, i.e., imagined items that users may purchase. The *trajectory manager* controls the granularity of imagined trajectories using the planning strategies, which balances the long-term rewards and short-term rewards. To optimize the action policy, the *imagination-augmented executor* minimizes the intrinsic imagination error of simulated trajectories by self-supervised reconstruction, while maximizing the extrinsic reward using model-free algorithms. Empirically, IRN promotes quicker adaptation to user interest, and shows improved robustness to the cold-start scenario and ultimately higher purchase performance compared to several baselines. Somewhat surprisingly, IRN using only the purchase reward achieves excellent next-click prediction performance, demonstrating that the agent can "guess what you like" via internal planning.

## 1 INTRODUCTION

A good recommender system can enhance both satisfaction for users and profit for content providers (Gomez-Uribe & Hunt, 2016). In many real-world scenarios, the recommender systems make recommendations based only on the current browsing session, given the absence of user profiles (because the user is new or not tracked or not logged in, till the final purchase step). A session is a group of sequential interactions between a user and the system within a short period of time. To model this phenomenon, Recurrent Neural Networks (RNNs) were recently employed as session-based recommenders (Hidasi et al., 2016; Jannach & Ludewig, 2017). For instance, GRU4Rec (Hidasi et al., 2016) utilizes the session-parallel mini-batch training to handle the variable lengths of sessions, and predicts the next action given the sequence of items in the current session. However, these approaches primarily focus on next-click prediction and model the session data via sequential classification, and thus cannot distinguish the different effects of user clicks and purchases.

In this paper, we consider the session-based recommendation as a Markov Decision Process (MDP), which can take into account both the click reward and the purchase reward (see Figure 1), and leverage Reinforcement Learning (RL) to learn the recommendation strategy. In practice, several challenges need to be addressed. First, the recommender systems involve large numbers of discrete actions (i.e., items), making current RL algorithms difficult to apply (Dulac-Arnold et al., 2015; Sunehag et al., 2015). This requires the agent to explore its environment for action feature learning and develop an ability to generalize over unseen actions. Second, we found it difficult to specify the click reward and the purchase reward; the policy may be biased by long sessions that contain many user clicks, as RL algorithms maximize the accumulated reward. Besides, real-world recommender systems require quick adaptation to user interest and robustness to the cold-start scenario (i.e., enhancing the purchase performance of short sessions). Therefore, we will be particularly interested in a case where only the purchase is used as reward (click sequences are used as inputs of the *imagination core* for

exploration).[1] However, the purchase reward is delayed and sparse (one session may contain only one purchase), making it a difficult signal for policy learning.

To augment reward and encourage exploration, we present the Imagination Reconstruction Network (IRN), which is inspired by the prediction error minimization (PEM) (Hohwy, 2016; Friston, 2010; Lotter et al., 2017) and embodied cognition (Clark, 2013; Burr & Jones, 2016; Seth, 2014; de Bruin & Michael, 2017) from the neuroscience literature. The PEM is an increasingly influential theory that stresses the importance of brain-body-world interactions in cognitive processes, involving perception, action and learning. In particular, IRN can be regarded as a proof-of-concept for the PEM from the recommendation perspective, following the ideas in Burr & Jones (2016) and Seth (2014) — the brain utilizes active sensorimotor predictions (or counterfactual predictions) to represent states of affairs in the world in an action-oriented manner. Specifically, the *imagination core* of IRN that predicts the future trajectories (i.e., a set of imagined items that user may purchase) conditioned on actions sampled from the imagination policy, can be considered as the generative model of the brain that simulates sensorimotor predictions. To update the action policy, the *imagination-augmented executor* minimizes the intrinsic imagination error of predicted trajectories by self-supervised reconstruction, while maximizing the extrinsic reward using RL, with shared input state or output action representations for predictive learning. This simulates the active perception (a key aspect of embodied cognition) of the body under the PEM framework, which adapts the agent to possible changes that arise from the ongoing exploratory action. Note that the imagination policy imitates the action policy through distillation or a delayed target network, and thus IRN constructs a loop between brain and body, encouraging the agent to perform actions that can reduce the error in the agent's ability to predict the future events (Pathak et al., 2017). IRN equips the agent with a planning module, *trajectory manager*, that controls the granularity of imagined trajectories using the planning strategies (e.g., breadth-$n$ and depth-$m$). Besides, IRN is a combination of model-based planning and self-supervised RL, as the imagined trajectories provide dense training signals for auxiliary task learning (see section 2).

The key contributions of this paper are summarized as follows:

- We formulate the session-based recommendation as a MDP, and leverage deep RL to learn the optimal recommendation policy, and also discuss several challenges when RL is applied.

- We consider a special case where only the purchase is used as reward, and then propose the IRN architecture to optimize the sparser but more business-critical purchase signals, which draws inspiration from the theories of cognition science.

- We present a self-supervised reconstruction method for predictive learning, which minimizes the imagination error of simulated trajectories over time. IRN achieves excellent click and purchase performance even without any external reward (predictive perception (Seth, 2014)).

- We conduct a comprehensive set of experiments to demonstrate the effectiveness of IRN. Compared to several baselines, IRN improves data efficiency, promotes quicker adaptation to user interest, and shows improved robustness to the cold-start scenario and ultimately higher purchase performance. These are highly valuable properties in an industrial context.

## 2 RELATED WORK

**Session-based Recommenders** Classical latent factor models (e.g., matrix factorization) break down in the session-based setting, given the absence of user profiles. A natural solution is the neighborhood approach like item-to-item recommendation (Mirowski et al., 2016). In this setting, an item similarity matrix can be precomputed based on co-occurrences of clicked items in sessions. However, this method only considers the last clicked item of the browsing session for recommendations, ignoring the sequential information of the previous events. Previous works also attempt to apply MDPs in the recommendation systems (Shani et al., 2002; Tavakol & Brefeld, 2014). The main issue is that the state space quickly becomes unmanageable due to the large number of items (IRN employs deep learning to overcome this problem and thus generalizes well to unseen states). Recently, RNNs have been used with success in this area (Hidasi et al., 2016; Hidasi & Karatzoglou, 2017; Jannach & Ludewig, 2017). GRU4Rec (Hidasi et al., 2016) is the first application of RNNs to model the session data, which can provide recommendations after each click for new sessions.

---

[1]We also conduct experiments that take into account the click reward.

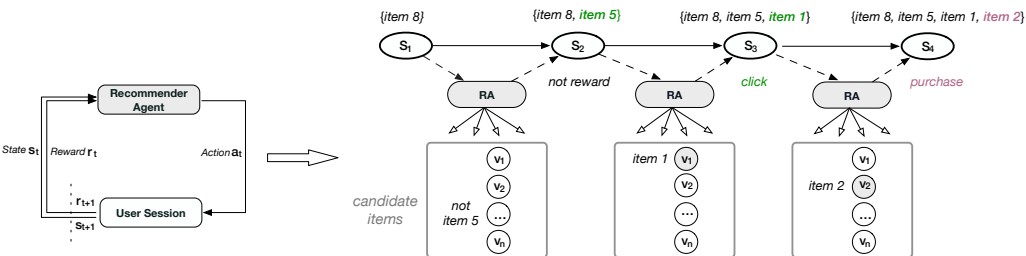

Figure 1: The agent-user interactions in MDP: the recommender agent (RA) generates a list of candidate items after each user click (green) or purchase (red).

However, GRU4Rec utilizes the session-parallel mini-batch training to handle the variable lengths of sessions; this trick cannot effectively capture sequentiality of sessions, since the network is trained using the BP algorithm (not BPTT for RNNs). These models primarily focus on next-click prediction and model the click-streams via sequential classification, while here we aim at modeling the purchase behavior and enhancing users' purchase intentions. Besides, IRN is built on RL, which encodes sequentiality of states into the value function.

**Imagination-augmented Agents**   All approaches incorporating off-policy experience (e.g., imagined trajectories) generated by a learned model can be categorized into model-based reinforcement learning (Racanière et al., 2017; Pascanu et al., 2017; Silver et al., 2017; Sutton, 1991). By using an internal model of the world, the agent can generalize to unseen states, remain valid in the real environment, and exploit additional training signals to improve data efficiency. However, the performance of model-based agents usually suffers from model errors resulting from function approximation. I2As (Racanière et al., 2017) were proposed to address this issue. I2As augment model-free agents with imagination and use an interpretation module to handle imperfect predictions. The imagined trajectories of I2As are provided as additional context (i.e., input features) to a policy network, while the proposed IRN uses the trajectories as additional training signals for self-supervised reconstruction.

**Self-supervised Reinforcement Learning**   In many real-world scenarios, reward is extremely sparse and delayed, and the agent updates its policy only if it reaches a pre-defined goal state. To model this phenomenon, self-supervised reinforcement learning have often been used, which accelerates the acquisition of a useful representation with auxiliary task learning (Jaderberg et al., 2016; Pathak et al., 2017; Mirowski et al., 2016; Shelhamer et al., 2016). Specifically, auxiliary tasks provide additional losses for feature learning, and can be trained instantaneously using the self-supervision from the environment. For instance, UNREAL (Jaderberg et al., 2016) maximizes many other pseudo-reward functions simultaneously, e.g., pixel change control, with a common representation shared by all tasks. In contrast, the proposed IRN do not require the external supervision from the environment, i.e., self-supervised reconstruction is performed on internal imagined trajectories.

## 3   PRELIMINARIES

We interpret the sequential recommendation task based on the standard reinforcement learning setting: An recommender agent (RA) interacts with an environment $\mathcal{E}$ (or user sessions) by sequentially choosing a list of recommendation items over a number of discrete time steps, so as to maximize its cumulative reward. As shown in Figure 1, we model this problem as a Markov Decision Process (MDP), which consists of a tuple of five elements $(S, A, P, R, \gamma)$:

*State space $S$*: A state $s_t \in S$ is defined as the previous items that a user clicked/purchased in one session. Specifically, the initial state $s_1$ contains the first item $i_0$ of one session. The items in $s_t = \{i_0, i_1, ..., i_{t-1}\}$ are sorted in chronological order.

*Action space $A$*: An action $a_t \in A$ is to recommend items to a user at time $t$ according to its policy $\pi$, where $\pi$ is a mapping from $s_t$ to $a_t$. We assume that the RA only recommends one item to the user each time, since we use the observed click/purchase sequences for off-policy training. During off-policy evaluation, we can recommend a list of $K$ candidates to the user.

*Reward $R$*: After the RA takes an action $a_t$ at the state $s_t$ , i.e., recommending an item to a user, the user browses this item and provides her feedback (click or purchase). The agent receives a scalar reward $r(s_t, a_t)$ according to the user's feedback. We also define the $k$-step return starting from state $s_t$ as $G_{t,t+k}(s_t) = \sum_{j=t}^{t+k} \gamma^{j-t} r(s_j, a_j)$, where $\gamma \in [0, 1]$ is a discounting factor.

*Transition probability $P$*: Transition probability $p(s_{t+1}|s_t, a_t)$ defines the probability of state transition from $s_t$ to $s_{t+1}$ when the RA takes action $a_t$. In this case, the state transition is deterministic after taking the ground-true action $a_t = i_t$, i.e., $p(s_{t+1}|s_t, i_t) = 1$ and $s_{t+1} = s_t \cup \{i_t\}$.

The goal of the RA is to find an optimal policy $\pi^*$, such that $V^{\pi^*}(s_1) \geq V^{\pi}(s_1)$ for all policies $\pi$ and start state $s_1$, where $V^{\pi}(s_t)$ is the expected return for a state $s_t$ when following a policy $\pi$, i.e., $V^{\pi}(s_t) = \mathbb{E}_{s_{j>t} \sim S, a_{j>t} \sim \pi}[G_{t,\infty}(s_t)]$ (or $\mathbb{E}_{s \sim \pi}[G_{t,\infty}(s_t)]$ for simplicity).

**Asynchronous Advantage Actor-Critic.** This paper builds upon the A3C algorithm, an actor-critic approach that constructs a policy network $\pi(a|s; \theta)$ and a value function network $V(s; \theta_v)$, with all non-output layers shared (Mnih et al., 2016). The policy and the value function are adjusted towards the bootstrapped $k$-step return $G_{t,t+k}(s_t) + \gamma^{k+1} V(s_{t+k+1}; \theta_v)$, $\mathcal{L}_{A3C} = \mathcal{L}_\pi + \mathcal{L}_{VR}$, where $\mathcal{L}_\pi = -\mathbb{E}_{s \sim \pi}[G_{1,\infty}(s_1)]$ and $\mathcal{L}_{VR} = \mathbb{E}_{s \sim \pi}[A(s_t, a_t)]$. The advantage function $A(s_t, a_t)$ (Baird III, 1993) is computed as the difference of the bootstrapped $k$-step return and the current state value estimate:

$$A(s_t, a_t) = G_{t,t+k}(s_t) + \gamma^{k+1} V(s_{t+k+1}; \theta_v^-) - V(s_t; \theta_v), \tag{1}$$

where $\theta_v^-$ are the parameters of the previous target network. To increase the probability of rewarding actions, A3C applies an update $g(\theta)$ to the parameters $\theta$ using an unbiased estimation (Sutton et al., 2000):

$$g(\theta) = \nabla_\theta \log \pi(a_t|s_t; \theta) A(s_t, a_t). \tag{2}$$

The value function $V(s; \theta_v)$ is updated following the recursive definition of the Bellman Equation, $V(s_t; \theta_v) = \mathbb{E}_{s \sim \pi}[G_{t,t+k}(s_t) + \gamma^{k+1} V(s_{t+k+1}; \theta_v)]$. Then $g(\theta_v)$ is obtained by minimizing a squared error between the target return and the current value estimate:

$$g(\theta_v) = -A(s_t, a_t) \frac{\partial}{\partial \theta_v} V(s_t; \theta_v). \tag{3}$$

In A3C multiple agents interact in parallel, with multiple instances of the environment. The asynchronous execution accelerates and stabilizes learning. In practice, we combine A3C with the session-parallel mini-batches proposed in (Hidasi et al., 2016). Each instance of the agent interacts with multiple sessions simultaneously, gathering $M$ samples from different sessions at a time step. After $k$ steps, the agent updates its policy and value network according to Eq. (2)(3), using $k * M$ samples. This decorrelates updates between samples of one session in the instance level. Besides, to build the A3C agent, we employ an LSTM that jointly approximates both policy $\pi$ and value function $V$, given the one-hot vectors of previous items clicked/purchased as inputs.

## 4 IRN ARCHITECTURE

In this section we incorporate the imagination reconstruction module into the model-free agents (e.g., A3C) in order to enhance data efficiency, promote more robust learning and ultimately higher performance under the sparse extrinsic reward. Our IRN implements an imagination-augmented policy via three key components (Figure 2). The *imagination core (IC)* predicts the next time steps conditioned on actions sampled from the imagination policy $\hat{\pi}$. At a time step $t$, the *trajectory manager (TM)* determines how to roll out the *IC* under the planning strategy, and then produces imagined trajectories $\hat{\mathcal{T}}_1, \ldots, \hat{\mathcal{T}}_n$ of an observable world state $s_t$. Each trajectory $\hat{\mathcal{T}}_j$ is a sequence of items $\{\hat{i}_{j,t}, \hat{i}_{j,t+1}, \ldots\}$, that users may purchase (or click) from the current time $t$. The *Imagination-augmented Executor (IAE)* aggregates the internal data resulting from imagination and external rewarding data to update its action policy $\pi$. Specifically, the *IAE* optimizes the policy $\pi$ by maximizing the extrinsic reward while minimizing the intrinsic imagination error. In principle, IRN encourages exploration and learns predictive representations via imagination rollouts, which promotes quick adaptation to user interest and robustness to the cold-start scenario.

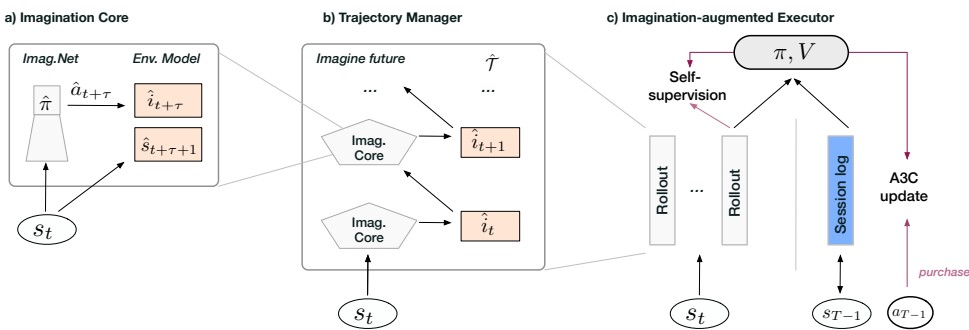

Figure 2: IRN architecture: a) the *imagination core (IC)* predicts the next time step and then generates the imagined trajectories $\hat{\mathcal{T}}$; b) the *trajectory manager (TM)* employs various planning strategies (e.g., depth-$m$ here) to control the granularity of $\hat{\mathcal{T}}$; c) the *imagination-augmented executor (IAE)* optimizes the network using the internal imagination data and external rewarding data (e.g., purchases).

**Imagination Core**   In order to simulate imagined trajectories, we rely on environment models that, given the present state and a candidate action, make predictions about the future states. In general, we can employ an environment model that build on action-conditional next-step predictors (Oh et al., 2015), and train it in an unsupervised fashion from agent experiences. However, the predictors usually suffer from model errors, resulting in poor agent performance, and require extra computational cost (e.g., pre-training). Besides, the predictors may learn a trivial identical function, since the state transition in agent trajectories (or session data) is deterministic, i.e., $s_{t+1} = s_t \cup \{i_t\}$ and $i_t = a_t$. In this work, we derive a static environment model from the state transition: $\hat{s}_{t+\tau+1} = \hat{s}_{t+\tau} \cup \{\hat{i}_{t+\tau}\}$, $\hat{i}_{t+\tau} = \hat{a}_{t+\tau}$ and $\hat{s}_t = s_t$, where $\tau$ is the length of the imagined rollout, $\hat{a}_{t+\tau}$ the output action of the imagination policy $\hat{\pi}$. During training, the generated item $\hat{i}_{t+\tau}$ may not be the true purchase/click, but we still use it for self-supervised reconstruction. This makes the action policy $\pi$ more robust to intrinsic errors and forces the imagination policy $\hat{\pi}$ to generate more accurate actions.

In practice, the imagination policy $\hat{\pi}$ can be obtained from policy distillation (Racanière et al., 2017) or a fixed target network like DQN (Mnih et al., 2015). The former distills the action policy $\pi(s_t; \theta)$ into a smaller rollout network $\hat{\pi}(s_t; \hat{\theta})$, using a cross-entropy loss, $l_{\pi,\hat{\pi}}(s_t) = \sum_a \pi(a|s_t) log \hat{\pi}(a|s_t; \hat{\theta})$. The latter uses a shared but slowly changing network $\hat{\pi}(s_t; \theta^-)$, where $\theta^-$ are previous parameters in $\pi(s_t; \theta)$. By imitating the action policy $\pi$, the imagined trajectories will be similar to agent experiences in the real environment; this also helps *IAE* learn predictive representations of rewarding states, and in turn should allow the easy learning of the action policy under the sparse reward signals.

**Trajectory Manager**   The *TM* rolls out the *IC* over multiple time steps into the future, generating multiple imagined trajectories with the present information. Additionally, various planning strategies are supported for trajectory simulation: breadth-$n$, depth-$m$ and their combination. For breadth-$n$ imagination, the *TM* generates $n$ trajectories, $\hat{\mathcal{T}}_1, \ldots, \hat{\mathcal{T}}_n$, over one time step from the current state $s_t$, i.e., $\hat{\mathcal{T}}_j = \{\hat{i}_{j,t}\}$. Empirically, the *IAE* using breadth-$n$ imagination will motivate the agent to focus on short-term events and predict the next step more accurately (e.g., enhancing the next-click prediction performance even when we do not formalize the click event as reward). For depth-$m$ imagination, the *TM* generates only one trajectory $\hat{\mathcal{T}}_1$ through $m$ time steps, i.e., $\hat{\mathcal{T}}_1 = \{\hat{i}_{1,t}, \ldots, \hat{i}_{1,t+m-1}\}$. This enables the agent to learn to plan the long-term future, and thus recommend items that yield high rewards (purchases). Finally, we can also achieve the trade-off between breadth-$n$ and depth-$m$ to balance the long-term rewards and short-term rewards. Specifically, we generate $n$ trajectories, and each has a depth $m$, i.e., $\{\hat{\mathcal{T}}\} = \{\{\hat{i}_{1,t}, \ldots, \hat{i}_{1,t+m-1}\}, \ldots, \{\hat{i}_{n,t}, \ldots, \hat{i}_{n,t+m-1}\}\}$.

**Imagination-augmented Executor**   As mentioned before, the *IAE* uses external rewarding data and internal imagined trajectories to update its action policy $\pi$. For the $j$-th trajectory, $\hat{\mathcal{T}}_j = \{\hat{i}_{j,t}, \ldots, \hat{i}_{j,t+m-1}\}$, we define a multi-step reconstruction objective using the mean squared error:

$$\mathcal{L}_j = \sum_{\tau=1}^{m} \gamma^{\tau} ||AE(\phi(\hat{\mathcal{T}}_{j,\tau})) - \phi(\hat{\mathcal{T}}_{j,\tau})||^2, \tag{4}$$

where $\hat{\mathcal{T}}_{j,\tau}$ is the $\tau$-th imagined item, $\phi(\cdot)$ is the input encoder shared by $\pi$ (for joint feature learning), $AE$ is the autoencoder that reconstructs the input feature, and the discounting factor $\gamma$ is used to mimic Bellman type operations. In practice, we found that action representation learning (i.e., the output weights of $\pi$) is crucial to the final performance due to the large size of candidate items. Therefore, we use the one-hot transformation as $\phi(\cdot)$ and replace $AE$ with the policy $\pi$ (excluding the final softmax function), and only back-propagate errors in the positions of imagined items. Specifically, for an imagined item, the mean squared error is computed between one and its activation value through $\pi$; errors for other items are turned to be zero. In this case, the policy $\pi$ is optimized not only to predict purchases accurately but also to minimize the reconstruction error of imagined items over time. Take a session for example, $\{i_0, i_1, ..., i_{q-1}, i_q\}$ ($i_q$ is the final purchased item), $\pi$ is trained $t + 1$ times using imagination reconstruction and once using A3C updating (for the purchase event); the overall reconstruction loss for this session is defined as $\mathcal{L}_{IRN} = \sum_{t=0}^{q} \sum_{j=1}^{n} \mathcal{L}_j(s_t)$.

There are several advantages associated with the imagination reconstruction. First, imagined trajectories provide auxiliary signals for reward augmentation. This speeds up policy learning when extrinsic reward is delayed and sparse. Second, by using a shared policy network, *IAE* enables exploration and exploitation, and thus improves feature learning when the number of actions is large. Third, compared with agents that predict the next observations for robust learning (Mirowski et al., 2016), our *IAE* reconstructs the imagined trajectories generated by the *TM* over time for predictive learning. When external reward is provided, *IAE* can be considered as a process of goal-oriented learning or semi-supervised learning. This self-supervised reconstruction approach also achieves excellent click and purchase prediction performance even without any external reward (unsupervised learning in this case, where inputs and output targets used for training $\pi$ are all counterfactual predictions, and the input states are transformed through actions in order to match predictions, i.e., predictive perception in Seth (2014)).

## 5 EXPERIMENTS

### 5.1 EXPERIMENTAL SETUP

We evaluate the proposed model on the dataset of ACM RecSys 2015 Challenge[2], which contains click-streams that sometimes end with purchase events. The purchase reward and the click reward (if used) are empirically set as 5 and 1, respectively. Focusing on the most recent events has shown to be effective (Jannach & Ludewig, 2017); therefore we collect the latest one month of data and keep sessions that contain purchases. We follow the preprocessing steps in Hidasi et al. (2016) and use the sessions of the last three day for testing (we also trained IRN and baselines on the full six month training set, with slightly poorer results; the relative improvements remained similar). The training set contains 72274 sessions of 683530 events, and the test set contains 7223 sessions of 63100 events, and the number of items is 9167. We also derive a separate validation set from the training set, with sessions of the last day in the training set. The evaluation is done by incrementally adding the previous observed event to the session and checking the rank of the next event. We adopt Recall and Mean Reciprocal Rank (MRR) for top-$K$ evaluations, and take the averaged scores over all events in the test set. We repeat this procedure 5 times and report the average performance. Without special mention, we set $K$ to 5 for both metrics. Besides, we build an environment using session-parallel mini-batches, where the agent interacts with multiple sessions simultaneously (see section 3).

**Baselines** We choose various baseline agents for comparison, including: (1) BPR (Rendle et al., 2009), a pairwise ranking approach, widely applied as a benchmark; (2) GRU4Rec (Hidasi et al., 2016), a RNN-based approach for session-based recommendations with a BPR-max loss function (note that original GRU4Rec gives much lower purchase performance, thus we only use the clicked items from the same mini-batch as negative examples); (3) CKNN (Jannach & Ludewig, 2017), a session-based KNN method, which incorporates heuristics to sample similar past sessions as neighbors; (4) A3C-F and A3C-P, the base agents without imagination, using the click and purchase

---

[2]http://2015.recsyschallenge.com/

Table 1: Recommendation performance for purchase and click events.

| Purchase | Recall@3 | Recall@5 | Recall@10 | MRR@3 | MRR@5 | MRR@10 |
|---|---|---|---|---|---|---|
| BPR | 0.471 | 0.679 | **0.820** | 0.372 | 0.434 | 0.458 |
| CKNN | **0.519** | **0.685** | 0.805 | **0.420** | **0.470** | **0.490** |
| GRU4Rec | 0.500 | 0.675 | 0.788 | 0.404 | 0.457 | 0.475 |
| A3C-F | 0.505 | 0.684 | 0.813 | 0.405 | 0.458 | 0.479 |
| A3C-P | 0.512 | 0.704 | 0.828 | 0.411 | 0.469 | 0.489 |
| IRN-F | 0.525 | 0.734 | 0.879 | 0.419 | 0.482 | 0.506 |
| PRN-P | 0.515 | 0.718 | 0.867 | 0.409 | 0.471 | 0.492 |
| IRN-P | **0.537** | **0.752** | **0.908** | **0.427** | **0.490** | **0.514** |
| **Click** | **Recall@3** | **Recall@5** | **Recall@10** | **MRR@3** | **MRR@5** | **MRR@10** |
| BPR | 0.198 | 0.249 | 0.276 | **0.168** | 0.182 | 0.187 |
| CKNN | **0.206** | 0.292 | 0.383 | 0.167 | 0.190 | 0.207 |
| GRU4Rec | 0.201 | **0.297** | **0.413** | 0.162 | **0.192** | **0.209** |
| A3C-F | 0.207 | **0.313** | **0.437** | 0.168 | 0.200 | **0.218** |
| A3C-P | 0.197 | 0.277 | 0.367 | 0.160 | 0.184 | 0.198 |
| IRN-F | 0.210 | 0.310 | 0.422 | 0.171 | 0.198 | 0.216 |
| PRN-P | 0.185 | 0.255 | 0.328 | 0.154 | 0.172 | 0.184 |
| IRN-P | **0.212** | 0.306 | 0.406 | **0.173** | **0.200** | 0.215 |

reward (-F) or only the purchase reward (-P); (5) IRN-F and IRN-P, the proposed models that augment A3C with imagiantion; (6) PRN-P, an A3C agent that reconstructs the previous observed trajectories (i.e., click/purchase sequences), using the purchase reward.

**Architecture**   We implemented IRN via Tensorflow[3], which will be released publicly upon acceptance. We use grid search to tune hyperparameters of IRN and compared baselines on the validation set. Specifically, the input state $s_t$ is passed through a LSTM with 256 units which takes in the one-hot representation of recent clicked/purchased items. The output of the LSTM layer is fed into two separate fully connected layers with linear projections, to predict the value function and the action. A softmax layer is added on top of the action output to generate the probability of 9167 actions. The discounting value $\gamma$ is 0.99. The imagination policy $\hat{\pi}$ is obtained from $\pi$ using the fixed target network, and the weights of $\hat{\pi}$ are updated after every 500 iterations. Without special mentioned, *TM* employs the combination of breadth-2 and width-2 for internal planning. The imagination reconstruction is performed every one environment step. The A3C updating is performed with immediate purchase reward (when found) or 3-step returns (when click reward is used). Besides, weights of IRN are initialized using Xavier-initializer (Glorot & Bengio, 2010) and trained via Adam optimizer (Kingma & Ba, 2014) with the learning rate and the batch size set to 0.001 and 128, respectively.

## 5.2   RESULTS

We first evaluate the top-$K$ recommendation performance. The experimental results are summarized in Table 1. From the purchase performance comparison, we get:

- A3C-P has already outperformed classical session-based recommenders (BPR, CKNN and GRU4Rec) on Recall metrics and achieved comparable results on MRR metrics. GRU4Rec gives poor purchase performance, as it focuses on next-click prediction.

- Comparing IRN-P with A3C-P, we can see that the purchase (and click) performance can be significantly improved with imagination reconstruction, demonstrating that IRN-P can guess what you like via internal planning and learn predictive representations.

- IRN-P consistently outperforms IRN-F, and A3C-P also outperforms A3C-F for purchase prediction. This demonstrates that purchase events can better characterize user interest, and the agents may be biased if clicks are used as reward.

---

[3]https://www.tensorflow.org

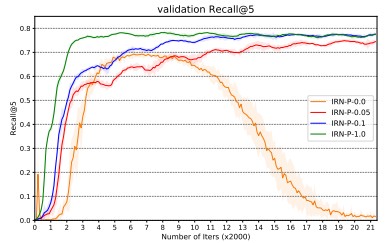
(a) Purchase *Recall@5* on the validation set

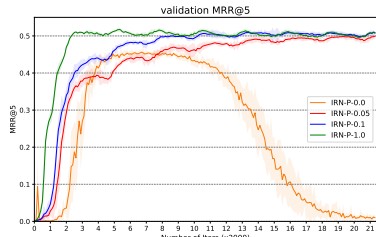
(b) Purchase *MRR@5* on the validation set

Figure 3: Possibility of being stuck in an non-optimal policy with varying reward sparsity for IRN.

Table 2: Purchase performance comparison with varying reward density $d$.

| | **Recall@5** | | | **MRR@5** | | |
|---|---|---|---|---|---|---|
| **Algorithm** | $d = 0.1$ | $d = 0.05$ | $d = 0.0$ | $d = 0.1$ | $d = 0.05$ | $d = 0.0$ |
| A3C-F | 0.679 | 0.676 | 0.674 | 0.460 | 0.459 | 0.459 |
| A3C-P | 0.687 | 0.578 | – | 0.464 | 0.398 | – |
| IRN-F | 0.730 | 0.726 | 0.720 | 0.480 | 0.478 | 0.473 |
| IRN-P | 0.735 | 0.725 | 0.653 | 0.482 | 0.475 | 0.432 |

- Comparing PRN-P with A3C-P and IRN-P, we found that reconstructing the previous actual trajectories (i.e., click-streams) also improves the purchase performance (compared to A3C-P). This is because that PRN-P can learn better representations for clicked items, and user purchases are sometimes contained in the click-streams. Besides, IRN-P outperforms PRN-P, since PRN-P introduces stronger supervision and may not know what is the final goal, while the imagination reconstruction (without any real trajectories) performs semi-supervised learning, which promotes more robust policy learning.

From the click performance comparison, we get:

- GRU4Rec achieves excellent next-click performance (e.g., top-5 and top-10) compared to BPR and CKNN, as it models the session data via sequential classification.

- A3C-F performs much better than A3C-P and GRU4Rec. This indicates that RL-based recommenders trained on clicks can generate actions that better preserve the sequential property, possibly due to the accumulated click reward (of longer sessions).

- Somewhat interesting, IRN-P significantly outperforms A3C-P, and gets comparable results like IRN-F and A3C-F. This demonstrates that the IRN-P agent may learn to plan and reconstruct the previous clicked trajectories even when only the purchase reward is provided.

**Varying the degree of purchase reward sparsity** We now explore the robustness of four RL-based recommenders to different purchase reward density. We randomly sample a $d$ proportion of purchase events from the training set. The click events remain unchanged. As shown in Table 2, A3C-F and IRN-F are robust to different purchase sparsity, since purchases are sometimes contained in the click sequences. IRN using only the click reward for policy learning can also enhance the purchase prediction performance (see $d = 0$). While the performance of A3C-P degrades with sparser purchase reward, the proposed IRN-P achieves comparable performance; the imagination reconstruction promotes predictive learning of rewarding states. To our surprise, we have found that IRN-P performs well even without any external reward from the environment (i.e., predictive perception, see A3C-F and IRN-P with $d = 0$). Minimizing the imagination error of predictive trajectories over time enables the agent to learn sequential patterns in an unsupervised fashion. Figure 3 compares the performance of IRN-P on different reward sparsity setting, where one epoch contains nearly 5000 iterations. We can observe that the performance of all models is gradually improved, and IRN-P with a larger $d$ learns faster, indicating better exploration and exploitation. Note that IRN-P with $d = 0$ will adversely decrease the performance due to the local over-training. In extreme cases,

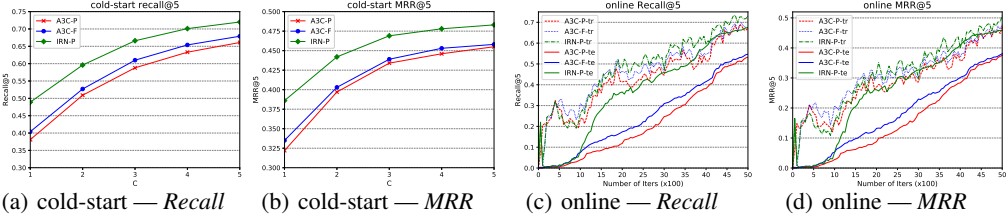

Figure 4: Purchase performance comparison on Recall@5 and MRR@5 metrics. (a, b) Results under the cold-start scenarios. (c, d) Results in online learning.

Table 3: Performance of IRN-P with different planning strategies (breadth-$n$ and depth-$m$).

| **n, m** | **Recall@5** | | | **MRR@5** | | |
|---|---|---|---|---|---|---|
| | First | Click | Purchase | First | Click | Purchase |
| 1, 1 | 0.397 | 0.307 | 0.733 | 0.299 | 0.199 | 0.477 |
| 2, 1 | **0.413** | **0.316** | 0.736 | **0.308** | **0.205** | 0.479 |
| 1, 2 | 0.344 | 0.294 | **0.755** | 0.280 | 0.190 | 0.488 |
| 2, 2 | 0.372 | 0.306 | 0.752 | 0.290 | 0.200 | **0.490** |

a final purchase decision would be unknown, the imagination reconstruction may be applied without external reward, but we can use the click prediction performance for validation and early stopping.

**Effectiveness of the trajectory manager** We then analyze the effectiveness of different planners of the *TM*. Table 3 shows the best results obtained with IRN-P when using alternative measurements. Note that the purchase event in one session is usually the last user interaction, and "First" means that the second event is evaluated separately (the first clicked item is used as the initial state). We can observe that, different planners equip the agent with different prediction capacity. For instance, IRN-P with a larger $n$ performs better on First and Click metrics, indicating that the agent with breadth-$n$ planning focuses more on short-term rewards. On the contrary, a larger $m$ can improve the purchase performance at a cost of lower First and Click results, since depth-$m$ planning enables the agent to imagine the longer future. The combination of breadth-$n$ and depth-$m$ can better balance the long-term rewards and short-term rewards. Besides, for IRN-P without any external reward ($d = 0.0$), the depth-2 planner gives better performance than depth-1 and breadth-2 on three measurements (by 2-5%), possibly due to the more predictive representations learned after unsupervised training. However, for IRN with purchase reward (semi-supervised learning), the purchase performance cannot be improved using longer imagined trajectories. One possible reason is that two steps of imagination reconstruction is sufficient for learning to predict the future events recursively; the first step of IRN learns to capture the difference of adjacent input states, and the second step learns to look ahead the future purchase signal accurately.

**Robustness to the cold-start scenario** We simulate a cold-start scenario using the test set. Specifically, we use a parameter $c$ to control the number of items in the input state (a set of one-hot vectors of clicked items), i.e., new events will not be added to the input state if the number of items exceeds $c$, but are still used for evaluations. Figure 4 (a,b) shows the purchase performance w.r.t. the cold-start parameter $c$. We can see that IRN-P outperforms A3C-P and A3C-F over all ranges of $c$, verifying the effectiveness of imagination reconstruction. In other words, IRN-P can guess what you like (or learn predictive representations) and obtain a better user (or session) profile. Besides, A3C-F achieves slightly better results than A3C-P, which is different from that in Table 1. A3C-F that trained with the click reward can preserve the sequential property of sessions, and thus provide auxiliary (implicit) information under the cold-start setting (in the warm-start setting, the agent using more clicked items as input may be biased and thus focuses on next-click prediction).

**Adaptation to user interest** To demonstrate that IRN can improve data efficiency and promote quick adaptation to user interest, we create a more realistic scenario for online learning. Specifically,

the training set is sorted in chronological order, and each event is used only once for training. The test set remains unchanged. Figure 4 (c,d) shows the purchase performance that is evaluated on both the training set ("-tr", averaged over a batch of training purchases) and the test set ("-te", averaged over all test purchases); for a given purchase event in the training set, the model first checks its rank and then uses it for an incremental update (measuring the short-term interest). We can see that, IRN-P promotes quick adaptation to user interest (after 2000 iterations) compared to A3C-P and A3C-F on the two datasets, and IRN-P-te shows improved data efficiency and purchase performance compared to A3C-F-te and A3C-P-te after online learning. Different from IRN-P, A3C-F and A3C-P perform poorly on the test set (compared to that of the last training batch); this highlights the importance of most recent events and demonstrates that IRN-P can capture user's long-term interest ahead of time.

## 6 CONCLUSION

In this paper, we propose the IRN architecture for session-based recommendation, which is inspired by the theories of cognition science. IRN can be regarded as a combination of model-based planning and self-supervised reinforcement learning, which employs a self-supervised reconstruction method for predictive learning, using the imagined trajectories generated by the internal model. We conducted experiments to study the impacts of difference components under different scenarios, verifying the effectiveness of our IRN architecture. We believe this kind of approaches has the potential to make a shift in the way we use recommender systems.

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
