# OpenReview forum: "Purchase as Reward : Session-based  Recommendation by Imagination Reconstruction"
_ICLR.cc/2019/Conference_

### Official Review · AnonReviewer3 · 2018-10-26
**I consider the proposed method interesting, although it is somewhat incremental. There are some conceptual issues with the proposed approach as well as missing related work. Motivation could be improved. The empirical evaluation is strong.**

**Rating:** 5
**Confidence:** 5

**Review:**

Summary:

The paper presents a session-based recommendation approach by focusing on user purchases instead of clicks. The method is inspired from concepts of cognitive science which adds an imagination reconstruction network to an actor-critic RL framework in order to encourage exploration.


Comments:

The proposed architecture is an interesting inspiration from Neuroscience which fits into the sequential recommendation problem. However, the motivation of using RL is missing from the technical contribution. Considering a deterministic policy, using LSTMs which already encode sequentiality of states in addition to another component for planning, seem to undermine the role of RL.

The motivation of creating imagined trajectories instead of actual user trajectories is unclear. On the other hand, there are many traditional planning approaches which are not mentioned such as Monte Carlo Tree Search that simultaneously trade-off exploration and exploitation.

The literature review is incomplete and misses important contributions on session-based recommendation, particularly, MDP-based methods such as Shani et al., An MDP-based recommender system, 2005 and Tavakol and Brefeld, Factored MDPs for Detecting Topics of User Sessions, 2014 (also see references therein).

Empirically, the authors compare their method to several recent baselines. This renders the empirical part exceptionally strong. Nevertheless, the length of the trajectories is only 2 and instead should be varied empirically to show the usefulness of the reconstruction network.


Questions:

-How are cold-start situations encountered if items are one-hot encoded?
-Why is there a strong focus on quick adaptation to user sessions? Usually, users tend to search quite a lot before converging; hence, longer sessions possibly better reflect user interests.


Minor:

-Proofreading is necessary
-Table 1 and 2 would be more readable if they were figures
-Figure 3 seems to be taken from Tensorflow runtime convergence plots, which could be dropped given the limited space

---

> ### Author Response · Authors · 2018-11-26
> **Thank you for your detailed reviews. We address your concerns below.**
>
> Q1: The motivation of using RL is missing from the technical contribution. Using LSTMs which already encode sequentiality of states in addition to another component for planning, seem to undermine the role of RL.
> A1: Similar to GRU4Rec, we also employ the session-parallel mini-batch trick to handle the variable lengths of sessions when training the LSTM. This training trick takes the current clicked item and the previous  (precomputed) hidden state as inputs of the RNN. As a result, GRU4Rec do not explicitly capture the sequential property of sessions, since the network is trained using the BP algorithm (not BPTT for RNNs). Details can be found in the author’s paper (Session-based recommendations with recurrent neural networks, Hidasi et al., 2016) and implementation (https://github.com/hidasib/GRU4Rec). Therefore, we utilize RL to alleviate this problem by encoding sequentiality of states into the value function. That’s why A3C-F performs better than GRU4Rec. Indeed, integrating the imagination reconstruction module into GRU4Rec also enhances the performance, but IRN with A3C achieves the best performance.
>
> Q2: The literature review is incomplete and misses important contributions on session-based recommendation, particularly, MDP-based methods.
> A2: We have added the review on MDP-based recommenders, including An MDP-based recommender system, Shani et al., 2015 and Factored MDPs for detecting topics of user sessions, Tavakol and Brefeld, 2014.
>
> Q3: The motivation of creating imagined trajectories instead of actual user trajectories is unclear. On the other hand, there are many traditional planning approaches which are not mentioned such as Monte Carlo Tree Search.
> A3: The IRN architecture is inspired by the theories of cognition science. Minimizing the reconstruction error of imagined trajectories adapts the agent to possible changes that arise from the ongoing exploratory action. When external reward is provided, the imagination-augmented executor (IAE) can be considered as a process of goal-oriented learning or semi-supervised learning. Introducing actual user trajectories into A3C-P (i.e., PRN-P, Table 1) also improves the prediction performance, but the proposed IRN-P significantly outperforms PRN-P, since PRN-P introduces stronger supervision and may not know what is the final goal. Besides, we do not employ MCTS for E&E, since we do not have the environment simulator like that of AlphaGo; a small fraction of real trajectories (i.e., offline session logs) is not enough to learn a value function that can approximate most state values for back-propagation (one step of MCTS).
>
> Q4: The length of the trajectories is only 2 and instead should be varied empirically to show the usefulness of the reconstruction network.
> A4: For IRN without any external reward (unsupervised predictive learning), longer imagined trajectories (>2) can further enhance the prediction performance (by 1%). For IRN with purchase reward (semi-supervised learning), the purchase performance cannot be improved using longer imagined trajectories. One possible reason is that two steps of imagination reconstruction is sufficient for learning to predict the future events recursively; the first step of IRN learns to capture the difference of adjacent input states, and the second step learns to look ahead the future purchase signal accurately.
>
> Q5: Why is there a strong focus on quick adaptation to user sessions?
> A5: In Introduction, we highlight that real-world recommender systems require (1) quick adaptation to user interest; (2) robustness to the cold-start scenario. Enhancing the purchase performance of short sessions is another expression for (2) but not for (1). Although longer sessions better reflect user interests, they would bias the training process of RL algorithms, which is unfair to short ones with the smaller accumulated reward. That's why we are interested in the case where only the purchase is used as reward. The proposed IRN achieves (1) and (2) by learning predictive representations. For (1), figure 4 shows that IRN-P-te generalizes much better to the test set (after 2000 iterations of online learning) compared to A3C-F-te and A3C-P-te. This demonstrates that the imagination reconstruction can promote quick adaptation and capture user’s long-term interest ahead of time.
>
> Q6: How are cold-start situations encountered if items are one-hot encoded?
> A6: For a given session, the input of IRN is a set of one-hot vectors of clicked items, which are fed into different timestamps of LSTMs. The evaluation is done by sequentially adding the current clicked item to the set and calculating the next performance. For the cold-start situations, we only add the first |c| clicked items (or their corresponding one-hot vectors) to the set to evaluate the later purchase event, which is mentioned in section 5.2 (“Robustness to the cold-start scenario”).

---

### Official Review · AnonReviewer2 · 2018-11-02
**The main idea of the paper was very interesting, but the clarity of the paper needs to be improved significantly**

**Rating:** 6
**Confidence:** 2

**Review:**

The paper proposed a new framework for session-based recommendation system that can optimize for sparse and delayed signal like purchase. The proposed algorithm with an innovative IRN architecture was intriguing.

The writing of the paper was not very clear and pretty hard to follow. With this level of clarity, I don’t think it’s easy for other people to reproduce the results in this paper, especially in section 4, where I expect more details about the description of the proposed new architecture. Even though the author has promised to release their implementation upon acceptance, I still think the paper needs a major change to make the proposed algorithm more accessible and easier for reproduce.

Some examples:
What is L_A3C in “L = L_A3C + L_IRN” in the first paragraph of session 4? It looks like a loss from a previous paper, but it’s kind hard to track what it is exactly.

“where Tj,τ is the τ-th imagined item, φ(·) is the input encoder shared by π (for joint feature learning), AE is the autoencoder that reconstructs the input feature, and the discounting factor γ is used to mimic Bellman type operations. … Therefore, we use the one-hot transformation as φ(·) and replace AE with the policy π (excluding the final softmax function), and only back-propagate errors of non-zero entries.”
This seems one of the most important components of the proposed algorithm, but I found it’s very hard to understanding what is done here exactly.

Regardless the sketchy description of the algorithm, the empirical results look good, with comprehensive baseline methods for comparison. It’s interesting to see the comparison between different reward function. Maybe the author can also discuss on the impact of the new imagination module on the training time.

---

> ### Author Response · Authors · 2018-11-30
> **Thank you for your appreciation of our contributions. We apologize if you had a hard time reading this paper.**
>
> Q1: What is L_A3C in “L = L_A3C + L_IRN” in the first paragraph of session 4? It looks like a loss from a previous paper, but it’s kind hard to track what it is exactly.
> A1: Thanks. Based on your comment, we have added detailed descriptions for both L_A3C and L_IRN. The L_A3C loss function is defined in Section 3 (“Asynchronous Advantage Actor-Critic”), which maximizes the external accumulated reward via policy gradient and value regression.
>
> Q2: “…Therefore, we use the one-hot transformation as φ(·) and replace AE with the policy π (excluding the final softmax function), and only back-propagate errors of non-zero entries.” This seems one of the most important components of the proposed algorithm, but I found it’s very hard to understanding what is done here exactly.
> A2: We have revised some flaws and supplemented detailed descriptions in Section 4 (especially “Imagination-augmented Executor”).  For an input state s_t,  we use the imagination policy (or action policy \pi for simplicity) to generate several imagined trajectories (each is a sequence of imagined items). To train the action policy \pi, we introduce the loss function L_IRN and use these imagined trajectories as inputs and output targets to compute the backward error. Note that different imagined items are fed into different timestamps of the LSTM. For an imagined item (one time step), the mean squared error of Eq.4 is computed between its activation value through \pi (prediction without softmax) and one (target label); errors for other items are turned to be zero. In other words, only errors for imagined items are back-propagated to optimize the policy network. Combined with A3C, the action policy \pi is optimized not only to predict purchases accurately but also to minimize the reconstruction error of imagined items over time. Take a session for example, {i_0 , i_1 , ..., i_{q−1} , i_q } (i_q is the ﬁnal purchased item), the policy network \pi is trained t+1 times using imagination reconstruction and once using A3C updating (for the purchase event).
>
> Q3: It’s interesting to see the comparison between different reward function.
> A3: For A3C-F and IRN-F (even GRU4Rec with different label weights), increasing the value of purchase reward can slightly improve the purchase performance at the cost of lower click performance; increasing the value of click reward degrades the purchase performance, due to the purchase reward sparsity. For IRN-P, only the purchase reward is used as supervision, the imagination reconstruction can be regarded as goal-oriented learning or semi-supervised learning. As a result, different purchase values give similar performance for IRN-P.

---

### Official Review · AnonReviewer1 · 2018-11-09
**The motivations of applying reinforcement learning to recommendation systems are not very convinced. Theory contributions may not be very significant**

**Rating:** 5
**Confidence:** 3

**Review:**

The paper aimed at improving the performance of recommendation systems via reinforcement learning. The author proposed an Imagination Reconstruction Network for the recommendation task, which implements an imagination-augmented policy via three components: (1)  the imagination core (IC) that predicts the next time steps conditioned on actions sampled from an imagination policy; (2) the trajectory manager (TM) that determines how to roll out the IC under the planning strategy and produces a set of imagined item trajectories; (3) the imagination-augmented executor (IAE) that aggregates the internal data resulting from imagination and external rewarding data to update its action policy.

Strengths of the paper:
(1) The research problem that the performance of recommendation systems needs to be improved is of great value to be investigated, as recommendation systems play crucial role in people’s daily lives.
(2) Experiments were conducted on a publicly available dataset.
(3) Robustness to cold-start scenario was tested and evaluated in the experiments.

Weaknesses of the paper:
(1) The motivations of applying reinforcement learning techniques are not convinced to me. There are a lot of supervised learning algorithms to the task of recommendations. Why do the authors utilize reinforcement learning to the task but not other supervised learning techniques? Is it because reinforcement learning based methods work better than traditional machine learning based ones? The motivations of integrating A3C (Asynchronous Advantage Actor-Critic) but not other techniques into the proposed model are not convinced to me as well.
(2) State-of-the-art reinforcement learning algorithms were not taken into account for baselines in the experiments. As the proposed method is built based on reinforcement learning, it would be better if the authors could include state-of-the-art reinforcement learning algorithms as their baselines.
(3) Some details are missing, resulting in the fact that it is hard for other researchers to fully capture the mechanism of the proposed algorithm. In equations (2) and (3), what is theta_v? How is theta_v associated with the parameters in LSTM. Is theta_v denoted the parameters of LSTM? How do the authors define the loss functions, i.e., \mathcal{L}_{A3C} and \mathcal{L}_{IRN}? What are the relationships among \mathcal{L}_{A3C}, \mathcal{L}_{IRN} and the one defined in equation (4)?
(4) The contributions of the paper in terms of theory are somewhat not significant. It seems that the proposed algorithm is built based on and combined by existing algorithms such as A3C.

Minor comments:
(1) It would be better if the authors can test the proposed model on more datasets. There are many publicly available datasets for testing the performance of recommendation systems.
(2) Figure 2 is not straightforward. It would be better if the authors can draw the figure in other ways. (I am not sure if the authors have expressed the underlying ideas clearly with Figure 2).

---

> ### Author Response · Authors · 2018-11-26
> **Apologies for being unclear in some parts of our paper. We address your concerns below.**
>
> Q1: The contributions of the paper in terms of theory are somewhat not significant. It seems that the proposed algorithm is built based on and combined by existing algorithms such as A3C.
> A1: IRN with GRU4Rec also significantly enhances the prediction performance. We would like to emphasize that the contributions of the paper lies primarily in two points: 1) a new application, i.e., optimizing the sparse and delayed purchase signal directly may be more business-critical, which is different from traditional CTR (Click-Through Rate Prediction) techniques; 2) the proposed imagination reconstruction module that is able to learn predictive representations even without any external reward (when training the action policy \pi, inputs and output targets used for self-supervised reconstruction are all counterfactual predictions, i.e., the predictive items in imagined trajectories \hat{\mathcal{T}}_{j,\tau}).
>
> Q2: The motivations of applying reinforcement learning. Why do the authors utilize RL to the task but not other supervised learning techniques?
> A2: We aim to solve the sequential recommendation problem, which meets the setting of MDP: the recommendation agent provides a list of candidate items to the user at a given timestamp, according to that observed state. Supervised learning techniques like GRU4Rec utilizes the session-parallel mini-batch training to handle the variable lengths of sessions. This training trick takes the current clicked item and the previous  (precomputed) hidden state as inputs of the RNN. As a result, GRU4Rec do not explicitly capture the sequential property of sessions, since the network is trained using the BP algorithm (not BPTT for RNNs). On the other hand, reinforcement learning alleviates this problem by encoding sequentiality of states into the value function (A3C-F performs better than GRU4Rec). Indeed, integrating the imagination reconstruction module into GRU4Rec also enhances the performance, but IRN with A3C achieves the best performance.
>
> Q3:  As the proposed method is built based on reinforcement learning, it would be better if the authors could include state-of-the-art reinforcement learning algorithms as their baselines.
> A3: We do not include other RL algorithms as baselines, which cannot further enhance the prediction performance. Different from Atari games, we do not have the environment simulator for recommendation systems. Therefore,  only a small fraction of real trajectories (i.e., offline session logs) is available, making it difficult to learn a better value function; A3C is enough to achieve a competitive result.
>
> Q4: How do the authors define the loss functions, i.e., \mathcal{L}_{A3C} and \mathcal{L}_{IRN}? What are the relationships among \mathcal{L}_{A3C}, \mathcal{L}_{IRN} and the one defined in equation (4)?
> A4: In section 5 (“architecture”), we mentioned that the imagination reconstruction is performed every one environment step while the A3C updating is performed with immediate purchase reward (when found) or 3-step returns (when click reward is used). Take a session for example, {i_0, i_1, ..., i_{q-1},i_q} (here i_q is the purchased item), IRN-P is trained t+1 times using \mathcal{L}_{IRN} and one time using \mathcal{L}_{A3C} (for the final purchase).  Definitions of these loss functions are added to the revised paper (see Section 3, "Asynchronous Advantage Actor-Critic" and Section 4, "Imagination-augmented Executor").
>
> Q5: In equations (2) and (3), what is theta_v? How is theta_v associated with the parameters in LSTM. Is theta_v denoted the parameters of LSTM?
> A5: A3C is an actor-critic approach that constructs a policy network π(a|s; theta) and a value function network V (s; theta_v ), with all non-output layers shared. Specifically, the output of the LSTM layer is fed into two separate fully connected layers to predict the state value and the probability of all actions, which is mentioned in section 5.1 (“Architecture”) and section 3 (“Asynchronous Advantage Actor-Critic”). In other words, theta and theta_v share the same parameters except for separate weights of two output layers; the parameters of LSTM is the union set of theta and theta_v.
>
> Q6: It would be better if the authors can test the proposed model on more datasets.
> A6: Although there are many publicly available datasets for recommendation systems, we only found this ACM RecSys 2015 dataset that contains both click-streams and final purchase events.

---

### Meta-Review · Area_Chair1 · 2018-12-17
**important application area, not sufficiently placed in the context of prior work (both conceptually and empirically)**

**Confidence:** 4
**Recommendation:** Reject

**Metareview:**

This paper addresses the problem of recommendations within user sessions from a reinforcement learning perspective. The problem is naturally modeled as an RL problem, given its sequential nature and inherent uncertainty of any model over user preferences. The problem suffers from delayed and sparse rewards, which the authors propose to address using self-supervised prediction. The approach is empirically validated in a simulated setting, using data from the 2015 ACM RecSys Challenge.

The reviewers and AC note that the problem studied is an important application area where RL has high potential to improve over current research results and industry practice. The proposed idea is interesting, and the strong empirical evaluation on a publicly available data set is highlighted. R1 also commends the authors' decision to address the challenging cold-start problem.

The reviewers and AC also note several potential weaknesses. The choice of addressing the problem from a reinforcement learning perspective is not clearly motivated. This is needed, as many supervised learning (and other types) approaches to the problem exist. A performance comparison to current state-of-the-art RL baselines is missing. The proposed approach is related to both imagination augmented (I2A, Racaniere et al. 2017) and agents with auxiliary rewards (UNREAL, Jaderberg et al. 2016), but does not compare to either method. Neither does the related work section sufficiently clarify why the proposed approach is expected to improve over these prior approaches. A thorough comparison to these baselines in a real-world application like session-based recommendation would be a strong contribution in itself, but without the contributions of the paper are hard to assess. Reviewers also noted lack of clarity. Some concerns are addressed by the authors, but the consensus is that the paper would benefit from a major revision to clearly work out the method, as well as it's conceptual and empirical differences from existing reinforcement learning approaches. R3 mentions missing related work, some of which the authors include in the revision. The AC recommends also following up on references in cited papers to ensure a future revision of the paper is well placed in the context of prior work on recommender systems, especially when modeled as a reinforcement learning problem.

Overall, the paper was assessed as borderline by the reviewers. The ACs view is that there are too many concerns for acceptance at ICLR in the present form, and that the paper will benefit from a thorough revision.